# Phase-separating pyrenoid proteins form complexes in the dilute phase

Guanhua He [1], Trevor GrandPre [2,3], Hugh Wilson[4], Yaojun Zhang[3,5,6], Martin C. Jonikas [1,7✉], Ned S. Wingreen [1,4✉] & Quan Wang [4,8✉]

While most studies of biomolecular phase separation have focused on the condensed phase, relatively little is known about the dilute phase. Theory suggests that stable complexes form in the dilute phase of two-component phase-separating systems, impacting phase separation; however, these complexes have not been interrogated experimentally. We show that such complexes indeed exist, using an in vitro reconstitution system of a phase-separated organelle, the algal pyrenoid, consisting of purified proteins Rubisco and EPYC1. Applying fluorescence correlation spectroscopy (FCS) to measure diffusion coefficients, we found that complexes form in the dilute phase with or without condensates present. The majority of these complexes contain exactly one Rubisco molecule. Additionally, we developed a simple analytical model which recapitulates experimental findings and provides molecular insights into the dilute phase organization. Thus, our results demonstrate the existence of protein complexes in the dilute phase, which could play important roles in the stability, dynamics, and regulation of condensates.

[1] Department of Molecular Biology, Princeton University, Princeton, NJ 08544, USA. [2] Department of Physics, Princeton University, Princeton, NJ 08544, USA. [3] Center for the Physics of Biological Function, Princeton University, Princeton, NJ, USA. [4] Lewis-Sigler Institute for Integrative Genomics, Princeton University, Princeton, NJ 08544, USA. [5] Department of Physics and Astronomy, Johns Hopkins University, Baltimore, MD, USA. [6] Department of Biophysics, Johns Hopkins University, Baltimore, MD, USA. [7] Howard Hughes Medical Institute, Princeton University, Princeton, NJ 08544, USA. [8] Present address: Laboratory of Chemical Physics, National Institute of Diabetes and Digestive and Kidney Diseases, National Institutes of Health, Bethesda, MA 20892, USA. ✉email: mjonikas@princeton.edu; wingreen@princeton.edu; quan.wang@nih.gov

Recently, liquid-liquid phase separation was found to drive the assembly of many cellular compartments that lack membranes, also referred to as biomolecular condensates, including nucleoli[1] and P granules[2]. Various environmental factors have been shown to impact the phase separation of these condensates, including temperature[3] and ionic strength[4]. Condensate assembly is further regulated by the properties of the constituent biomolecules, such as multivalence of binding domains[5], the presence of intrinsically disordered regions[6], and post-translational modifications[7]. Recently, much effort has been devoted to studying the composition, material properties, and structure of condensates both in vivo[8,9] and in vitro[10]. However, relatively little is known about the molecular interactions outside of the condensates, i.e., in the dilute phase.

Recent computer simulations[11–13] of two-component systems suggest that condensate proteins form small complexes in the dilute phase and that the properties of these dilute-phase complexes play critical roles in regulating phase separation. In these theoretical studies, protein complexes were found to be prevalent in the dilute phase, lowering the dilute-phase free energy and thus shifting phase boundaries. Indeed, in some special 'magic-number' cases, the dilute phase was seen to be dominated by small complexes, typically dimers or trimers, that enjoy extra translational entropy with little or no cost in enthalpy, thus strongly competing with the dense phase and dramatically suppressing phase separation[11–13]. A similar model was proposed for protein-RNA systems, leading to a similar conclusion that stable oligomers in the dilute phase inhibit phase separation[14]. Taken together, these theoretical studies suggest that biomolecular interactions in the dilute phase could regulate phase separation. However, such complexes have not been directly observed in experiments in equilibrated liquid-liquid phase-separating systems.

In this work, we set out to characterize protein complexes in the dilute phase. We used an in vitro reconstitution system recapitulating the liquid-like pyrenoid matrix of the model alga *Chlamydomonas reinhardtii* (Fig. 1a). Prior studies showed that the pyrenoid is mainly composed of the $CO_2$-fixing enzyme, Ribulose-1,5-bisphosphate carboxylase/oxygenase (Rubisco), and its intrinsically disordered linker protein EPYC1[15]. Rubisco contains 8 small subunits and 8 large subunits. EPYC1 has 5 binding motifs that interact with Rubisco small subunits[16] (Fig. 1b), manifesting the classic 'stickers-and-spacers' architecture[17]. Upon mixing in vitro, purified Rubisco and EPYC1 proteins are both necessary and sufficient to drive phase separation[18] (Fig. 1c). Here, using Fluorescence Correlation Spectroscopy (FCS), we observed that EPYC1 and Rubisco form complexes in the dilute phase, both in the presence and absence of condensates. We found that the majority of these dilute phase protein complexes contain one Rubisco molecule. Further, we developed a theoretical model based on an analytical dimer-gel theory[13,19]. Our model generates a phase-diagram in good agreement with experiments, and successfully recapitulates the experimentally observed change in dilute phase composition as a function of Rubisco to EPYC1 ratio. This work demonstrates the existence of protein complexes in the dilute phase of a natural two-component phase-separating system, which paves the way to better understand how complexes in the dilute phase could regulate phase separation.

## Results

### Rubisco and EPYC1 phase separate in vitro over a broad range of protein concentrations.

To quantitatively measure the phase separation of Rubisco and EPYC1, we first obtained a phase diagram over a wide range of EPYC1 and Rubisco concentrations (Fig. 1d) using two complementary methods, a turbidity assay and fluorescence microscopy[20] (Methods and Supplementary Fig. 1). The turbidity assay monitors the droplet-induced light scattering at 340 nm as a proxy for phase separation and the fluorescence microscopy directly images the droplet formation via doping the system with 20 nM EPYC1-GFP. Both methods yielded similar results: we observed robust phase separation across a wide concentration range of EPYC1 and Rubisco (EPYC1:1-4 $\mu$M, Rubisco:0.05-1 $\mu$M). We did not observe phase separation with EPYC1 or Rubisco by itself (Supplementary Fig. 1a).

Based on this phase diagram, we designed FCS experiments that probe the dilute-phase composition: first, we characterized the dilute-phase complexes in a region of the phase diagram where phase separation does not occur, specifically 10 nM EPYC1 at varying Rubisco concentrations. Next, we characterized the dilute-phase composition in regions of the phase diagram where the dilute phase coexists with the condensed phase.

### Detecting EPYC1-Rubisco protein complexes by Fluorescence Correlation Spectroscopy.

We first set out to quantitatively measure EPYC1-Rubisco interactions. Interactions between EPYC1 and Rubisco had been studied using immunoprecipitation[15], yeast two-hybrid[21], and phase separation assays[18] (Fig. 1), but key quantitative information including the dissociation constant $K_d$ and complex composition is still lacking. Here, we used fluorescence correlation spectroscopy (FCS) to look for small EPYC1-Rubisco complexes in the dilute phase (Fig. 2).

FCS is a powerful biophysical technique to detect and quantify protein-protein interactions, both in solution[22] and in complex environments (e.g., in cells[23]). Importantly, the high spatial resolution of FCS (detection volume < 1 $\mu$m$^3$, Supplementary Fig. 2) and its high sensitivity (picomolar to nanomolar concentrations) enable the use of this technique to measure protein interactions in the dilute phase of a phase-separated system without disturbing the equilibrium between the dilute and dense phases.

Conceptually, FCS monitors the fluorescence intensity fluctuations arising from tagged protein molecules moving into and out of a small detection volume and uses time-correlation analysis to provide quantitative information on the target protein's concentration and diffusivity. The protein's diffusivity can be estimated by fitting the intensity autocorrelation curve, which depicts the correlation of the fluorescence signal with itself, shifted by various delay (or autocorrelation) times $\tau$. The formation of complexes by the target protein can be detected via a slowdown of the protein's measured diffusivity, which appears as a shift of the autocorrelation curve toward longer autocorrelation times.

To perform an FCS experiment, samples are placed inside a chamber with a bottom coverslip surface. In practice, we noted that EPYC1 protein has an unusually-high tendency to adsorb or nucleate on coverslip surfaces (Supplementary Fig. 2), even on surfaces receiving canonical modifications for single-molecule experiments, including polyethylene glycol (PEG), polyelectrolyte multilayer or detergent[24] (Supplementary Fig. 2). EPYC1 aggregation on these surfaces interfered with accurate correlation analysis. We, therefore, developed a new protocol based on electrostatically pre-coated polyethyleneimine-graft-polyethylene glycol (PEI-g-PEG, Methods) and found it to produce a surface that completely eliminates EPYC1 aggregation while alleviating (although not completely suppressing) Rubisco adsorption (Supplementary Fig. 2, Supplementary Note 1 and Methods). We performed all the imaging and FCS experiments using this improved protocol.

### EPYC1 and Rubisco form small complexes with a $K_d$~30 nM in solution.

We first measured the diffusion coefficients of EPYC1 alone and Rubisco alone. Knowing that an EPYC1 protein is much smaller (35 kDa) than a Rubisco holoenzyme (550 kDa), we expected that EPYC1 would diffuse more rapidly than Rubisco.

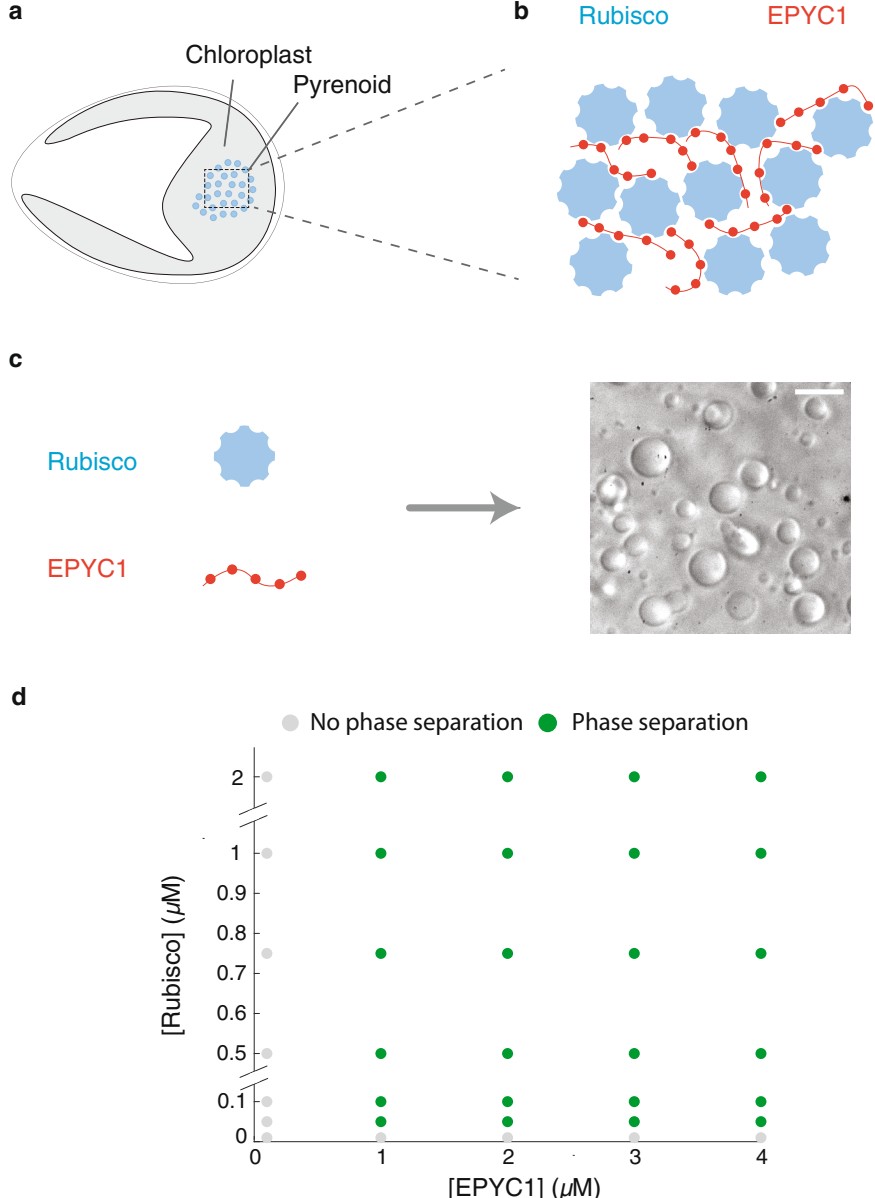

**Fig. 1 The components of the pyrenoid—EPYC1 and Rubisco—phase separate in vitro over a broad range of protein concentrations. a** Sketch of a *Chlamydomonas reinhardtii* cell, highlighting the chloroplast and the pyrenoid. The blue circles indicate Rubisco holoenzymes in the pyrenoid matrix. **b** Cartoon illustrating that the pyrenoid matrix is held together by multivalent interactions between EPYC1 and Rubisco. **c** Purified EPYC1 and Rubisco, when mixed together, phase separate in vitro. Scale bar, 5 $\mu$m. **d** Phase diagram of Rubisco-EPYC1 phase separation. Concentrations are expressed in terms of Rubisco holoenzymes and EPYC1 proteins.

Indeed, we found that EPYC1-GFP alone diffuses at ~62 $\mu$m$^2$/s while Rubisco-Alexa488 alone diffuses at ~38 $\mu$m$^2$/s (Fig. 2e).

For experiments mixing the two proteins together, we tagged EPYC1, reasoning that binding of Rubisco would lead to a greater observed change in diffusion rate than if we had tagged Rubisco and tried to detect binding of EPYC1 (Fig. 2a). We mixed a constant amount of EPYC1-GFP (10 nM) with varying concentrations of unlabeled Rubisco and recorded fluorescence intensity time traces (Fig. 2b). We found that when the Rubisco concentration is low (1 nM), the autocorrelation curve is indistinguishable from that of EPYC1-GFP by itself (Fig. 2d), suggesting the majority of EPYC1 remains unbound. On the other hand, when we introduced a higher concentration (50 nM) of Rubisco, the autocorrelation curve shifted to longer timescales, consistent with the hypothesis that EPYC1-Rubisco complexes are formed (Fig. 2c).

To quantify the EPYC1-Rubisco interaction, we extracted the EPYC1 diffusion coefficient from our FCS data (Methods) and plotted it against Rubisco concentration (Fig. 2e). We observed a gradual decrease of the EPYC1 diffusion rate as larger amounts of Rubisco were added to the solution. Note that since FCS measures average diffusion rates, the observed decay in diffusivity presumably indicates that the percentage of bound EPYC1 increases as more Rubisco is added, until the diffusion rate saturates at a plateau. We also showed that the decrease in the diffusion coefficient was not an artifact of Rubisco aggregation or of nonspecific GFP-Rubisco interaction (Supplementary Fig. 3). Further, a Rubisco mutant, previously shown to almost completely disrupt EPYC1-Rubisco interaction and pyrenoid formation in vivo[16], does not slow down the diffusion of EPYC1-GFP (Supplementary Fig. 4).

We attempted to extract the dissociation constant $K_d$ by fitting the data in Fig. 2e using a quadratic binding equation[25]

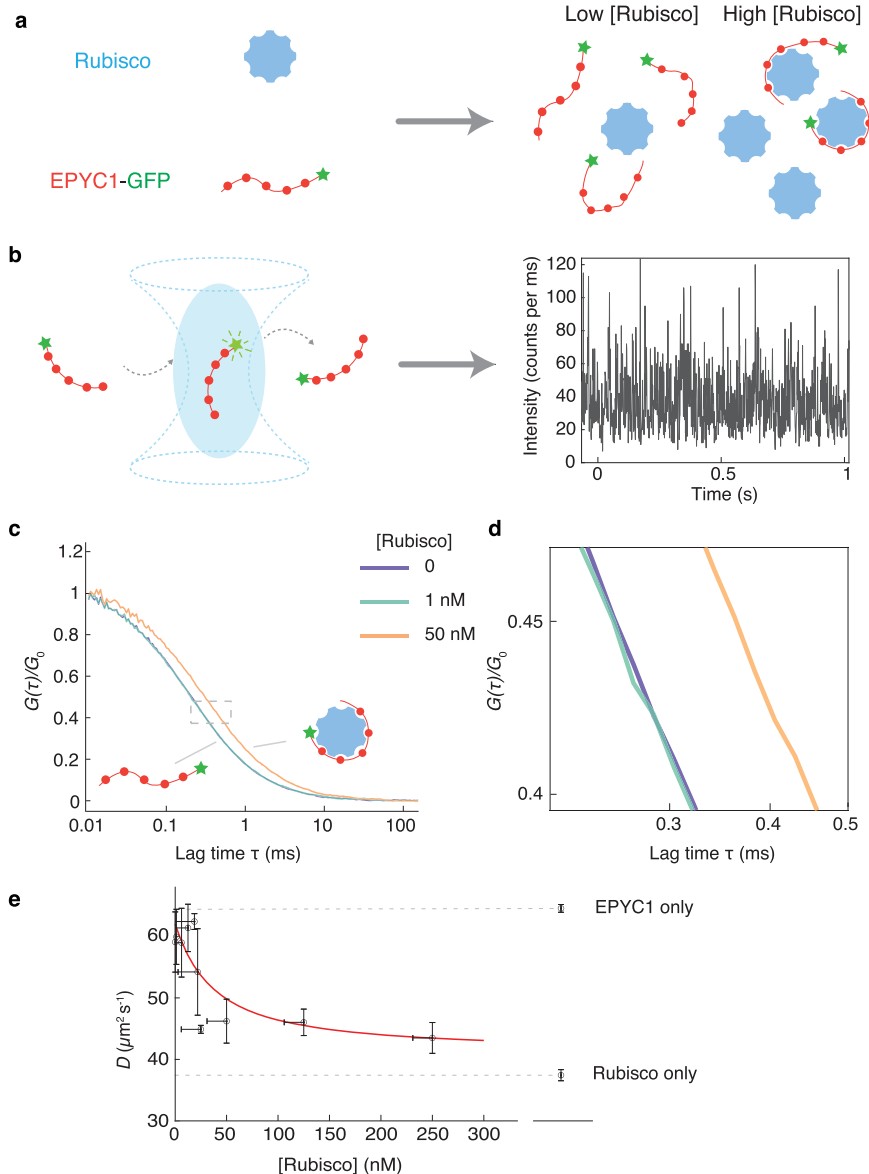

**Fig. 2 EPYC1 and Rubisco form complexes when mixed at low concentrations in vitro. a** Cartoon illustrating how EPYC1 and Rubisco coexist at different concentrations in the one-phase region. **b** Schematic depiction of FCS and an example of an EPYC1-GFP fluorescence intensity time trace ([EPYC1-GFP] = 10 nM). **c** Fluorescence intensity autocorrelation curves of 10 nM EPYC1-GFP fluorescence at three different Rubisco concentrations. Each autocorrelation curve is normalized by its fitted $G_0$ (e.g., correlation value at $\tau = 0$, Methods). **d** An enlarged version of the region shown with dashed lines in **c**. **e** Diffusion coefficient ($D$) of EPYC1-GFP inferred from FCS as a function of Rubisco concentration. Vertical error bars are standard deviations of repeated experiments ($n = 3$). The horizontal error bars indicate the estimated uncertainty of the Rubisco concentration in the solution due to protein loss in the measurement chamber (Supplementary Note. 1). The red curve is a fit to a quadratic model (Supplementary Note. 2) with a $K_d$ of ~30 nM. The diffusion rates for EPYC1-GFP-only and for Rubisco-Atto488-only are shown on the right. Concentrations are expressed in terms of Rubisco holoenzymes and EPYC1 proteins.

(Supplementary Note 2). The full quadratic equation is required because we cannot assume that only a small fraction of the Rubisco is bound to EPYC1. The resulting fit is consistent with simple A + B binding at equilibrium, yielding an estimated $K_d$ of $29 \pm 12$ nM (mean ± 68% confidence interval, Fig. 2e, red curve, Methods). Given that the measured $K_d$ is close to the fixed concentration of EPYC1 (10 nM), we consider this number to be an upper bound on the dissociation constant (Methods).

These FCS experiments also provide insight into the stoichiometry of the complexes that form in solution (without phase separation). At high Rubisco concentrations, the measured EPYC1-GFP diffusion coefficient reached a plateau at ~41 $\mu$m²/s, which is similar to the diffusion coefficient of Rubisco alone at ~38 $\mu$m²/s.

Therefore, it is likely that the majority of EPYC1-Rubisco complexes contain only one Rubisco (Fig. 2e) (we expect that complexes with two or three Rubiscos would diffuse roughly at 30 $\mu$m²/s and 26 $\mu$m²/s, respectively, assuming $D \propto (\text{MW})^{-1/3}$ for globular proteins[26], where $D$ is diffusion coefficient and MW is molecular weight).

**EPYC1 and Rubisco form complexes in the dilute phase alongside condensates.** The existence of EPYC1-Rubisco complexes in the one-phase region implies that such complexes should also exist in the dilute phase alongside condensates in the two-phase region. To test this hypothesis, we conducted FCS

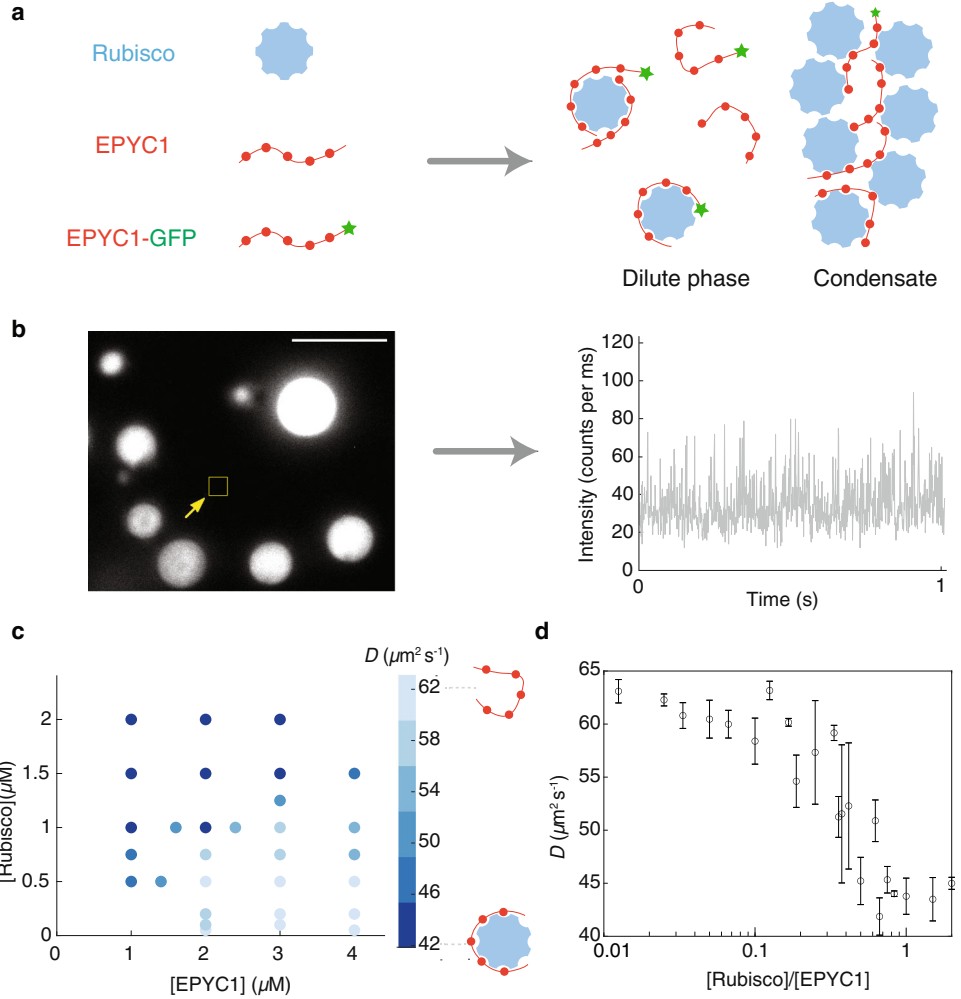

**Fig. 3 EPYC1 and Rubisco form complexes in the dilute phase of a phase-separated system. a** Cartoon depicting EPYC1 and Rubisco complexes in the dilute phase alongside a condensate. **b** FCS was performed in a focal volume away from condensates to measure the diffusion rate of EPYC1-GFP in the dilute phase. The EPYC1-GFP concentration was fixed at 20 nM. Left image represents typical field of view (bulk concentrations: [EPYC1] = 4 μM, [Rubisco] = 0.75 μM), scale bar = 5 μm. Right image is a representative fluorescence intensity time trace in the dilute phase. **c** Diffusion coefficients ($D$) of EPYC1-GFP in the dilute phase in the presence of condensates obtained by FCS. Obtained $D$ values are color coded as shades of blue (lighter blue represents higher $D$) and plotted on the phase diagram of Rubisco and EPYC1 concentrations. $D$ values are the average values of 3 repeat experiments. **d** The diffusion coefficients ($D$) in **c** plotted against overall Rubisco/EPYC1 concentration ratio. Error bars are standard deviations of repeated experiments. Note that because the x-axis is the [Rubisco]/[EPYC1] ratio, in some cases different data points in **c** are clustered into one data point in **d** (for example, $D$ values of 1 μM Rubisco with 2 μM EPYC1 and 0.5 μM Rubisco with 1 μM EPYC1 are summarized into one data point where the [Rubisco]/[EPYC1] ratio is 0.5). Concentrations are expressed in terms of Rubisco holoenzymes and EPYC1 proteins.

measurements in the dilute phase within a phase-separated mixture. We mixed EPYC1-GFP (fixed at 20 nM), unlabeled EPYC1 (from 1 to 4 μM), and Rubisco (ranging from 0.05 to 2 μM) to induce phase separation (Fig. 3). For each condition, we first verified droplet formation (Fig. 3b) under the microscope in wide-field imaging mode (Methods) and then aimed the laser at a region visually identified as the dilute phase and recorded fluorescence intensity time traces (Fig. 3b and Supplementary Fig. 5). For each raw intensity trace, we followed the same data analysis pipeline as the previous section, by calculating the autocorrelation function and extracting the EPYC1 diffusion coefficient from a fit. We plotted the EPYC1 diffusion coefficients as a 2D plot against the input concentrations of Rubisco and EPYC1 (Fig. 3c). We found that the measured EPYC1 diffusion coefficient, which we expect to be dependent on EPYC1 complex formation in the dilute phase and thus on the dilute-phase composition, depends strongly on the bulk EPYC1 and Rubisco concentrations. At overall high EPYC1 and low Rubisco concentrations (lower right

corner of the phase diagram), most of the EPYC1 in the dilute phase remained unbound (i.e., with a measured diffusion coefficient similar to EPYC1 alone at $D \sim 62$ μm$^2$/s), while at overall low EPYC1 and high Rubisco concentrations (upper left of the phase diagram), most dilute-phase EPYC1s were bound ($D \sim 42$ μm$^2$/s). To quantitatively understand how the bulk concentration ratios determine EPYC1 bound status in the dilute phase, we replotted the same dataset in Fig. 3d as EPYC1 diffusion coefficients against Rubisco/EPYC1 concentration ratio. We found that when the overall Rubisco/EPYC1 concentration ratio is low (between 0.01 and 0.3), EPYC1 has high diffusion rates, suggesting that the EPYC1s in the dilute phase are primarily free, unbound to Rubisco. (Note that the high EPYC1 diffusion rates in this regime do not mean that the Rubiscos in the dilute phase are unbound; it only means that there is more EPYC1 than Rubisco.) By contrast, for overall Rubisco/EPYC1 concentration ratios above 0.3, the EPYC1 diffusion rates slow down, implying that the dilute-phase EPYC1s are primarily in complexes with

Rubisco. Notably, the diffusion rates of bound EPYC1 in the dilute phase with condensates are very similar to those without condensates in Fig. 2. This similarity suggests that with or without condensates, the dilute-phase EPYC1-Rubisco complexes include only one Rubisco.

In conclusion, we found EPYC1 and Rubisco form complexes in the dilute phase in equilibrium with condensates over a broad range of Rubisco and EPYC1 concentrations. The fraction of EPYC1 in complexes depends on bulk Rubisco/EPYC1 concentration ratios. These results confirm the importance of dilute-phase complexes in shaping the overall phase diagram.

**Model for EPYC1-Rubisco phase separation and dilute-phase diffusion**. To gain a better understanding of EPYC1-Rubisco phase separation and the concentration-dependent EPYC1 diffusion coefficient in the dilute phase, we employed an analytical free-energy model adapted from dimer-gel theory[13,19] (Methods). EPYC1 and Rubisco are treated as polymers with, respectively, 5 stickers and 8 stickers which can form heterotypic sticker-sticker bonds. Inspired by the experimental findings here that dilute-phase protein complexes exist and only contain one Rubisco, we model the dilute phase as composed of Rubisco-EPYC1 dimers along with free Rubiscos and EPYC1s. We model the condensed phase as a gel of independent stickers. All proteins are also assumed to interact via excluded volume (that is, the space already occupied by one molecule cannot be occupied by other molecules), with the volume of Rubisco given by a sphere of diameter 10 nm[11] and the volume of EPYC1 given by its radius of gyration which is approximately 5 nm (Methods).

Our model uses the experimentally measured dissociation constant of EPYC1 and Rubisco, which from Fig. 2e is $K_d = 29 \pm 12$ nM (mean ± 65% CI), and then has only one fitting parameter which is the dissociation constant $K_b$ of independent stickers. We fit $K_b = 60\,\mu$M to best match the phase diagram measured by experiments. In practice, the precise value of $K_d$ is not critical to the overall agreement between model and data because changing $K_b$ and $K_d$ simultaneously while keeping $K_b^{c_b}/K_d^{\rho_d}$ (see the first terms on the right-hand side of Eqs. 10 and 11 in Materials and Methods) constant roughly leaves the phase diagram unchanged, where $\rho_d$ is the concentration of EPYC1-Rubisco dimers in the dilute phase and $c_b$ is the concentration of bound sticker pairs in the dense phase. However, the parameters $K_d$ and $K_b$ hardly change the dense-phase boundary which depends mainly on the excluded volume interactions. Despite its simplicity, our model is able to generate a phase diagram (Fig. 4a, b) that agrees well with the experimental dilute-phase boundary in Fig. 1d. Specifically, the model predicts that phase separation will occur even when EPYC1 concentrations are over 40 times larger than Rubisco concentrations, which agrees with experiment. The asymmetry of the model phase diagram arises from the difference in the number of stickers between EPYC1 and Rubisco as well as their different excluded volumes (Methods). The theoretical dense-phase boundary, which effectively has no fitting parameters, also agrees within a factor of 2–3 with a previous in vivo cryo-electron microscopy measurement of Rubisco density in the pyrenoid of ~600 $\mu$M[11], with some of the difference potentially arising from crowding in vivo.

Additionally, the model allows prediction of EPYC1 diffusion rates in the dilute phase. A zoomed in version of the phase diagram in Fig. 4b shows the experimental range of concentrations, with the black dots chosen to be close to the specific experimental concentrations used in Fig. 3c. For these specific overall concentrations, predictions for the dilute-phase concentrations of Rubisco and EPYC1 are obtained by following the tie

lines under the black dots to the dilute-phase boundary. The theory predicts the distribution of complexes that make up the dilute phase at every point on the dilute-phase boundary; in fact, since binding between EPYC1 and Rubisco is quite strong—the $K_d$ value of 30 nM being much smaller than the dilute-phase concentrations—essentially all dilute-phase proteins that can form dimers do so. Consequently, as illustrated by the boxes, where dilute-phase Rubisco concentrations are higher than EPYC1 concentrations, all EPYC1s will be bound to Rubiscos to form heterodimers leaving an excess of Rubiscos, and vice versa, where EPYC1 concentrations are higher than Rubisco concentrations all Rubiscos will form heterodimers, leaving an excess of EPYC1s. In Fig. 4c, the predicted fraction of dilute-phase EPYC1s that are left as monomers is plotted for points on the dilute-phase boundary corresponding to the black dots in Fig. 4b as a function of the overall Rubisco/ EPYC1 concentration ratio. As indicated by the right axis in Fig. 4c, the predictions for the EPYC1 monomer fraction can then be used to predict the EPYC1 diffusion coefficient, taken as the weighted average of the empirical monomer and dimer diffusion coefficients (respectively ~62 $\mu$m$^2$/s and ~42 $\mu$m$^2$/s). Our model nicely reproduces the steep decrease in the measured dilute-phase EPYC1 diffusion coefficient as a function of increasing overall Rubisco to EPYC1 concentrations (Fig. 3d) and clarifies that the sharp decrease is due to the drastic increase in the fraction of dilute-phase EPYC1 that form dimers with Rubisco.

## Discussion

In this study, we used fluorescence correlation spectroscopy (FCS) to probe the dilute phase organization of the EPYC1-Rubisco system in vitro, in both the one-phase and two-phase regions of the measured phase diagram. FCS has recently been used to probe various properties of biological condensates, including binodals[27,28] and viscosity in the dense phase[27]. Here, we use FCS to selectively probe the dilute phase and measure diffusivity of molecules (and molecular complexes) outside of condensates. We then link the measured diffusion coefficient to the presence of EPYC1-Rubisco complexes. Our study takes advantage of the large difference in molecular weight between EPYC1 (35 kDa) and Rubisco (550 kDa), so that Rubisco binding to EPYC1 produces a slowdown of EPYC1 diffusion (here by a factor of 1.7). We envision the method to be applicable to other systems undergoing phase separation, but care must be taken in the experimental design when the interacting partners have similar sizes[29] or when conducting the measurement in a cellular environment[23].

The main finding of this work is the direct experimental observation that EPYC1 and Rubisco form small complexes in both subsaturated solutions and in the dilute phase of a phase-separated, two-component mixture. These complexes seem to contain at most one Rubisco and are most likely EPYC1-Rubisco heterodimers (i.e., 1:1 stoichiometry). The fact that these complexes coexist with condensates suggests that they are true equilibrium species, as opposed to unstable intermediate oligomers formed during nucleation[30]. These stable complexes thus compete with condensates to shape the equilibrium phase diagram. Recently, Kar et al.[31] reported the existence of heterogeneous molecular clusters larger than ~100 nm in subsaturated solutions of FUS (and of other RNA-binding proteins with disordered domains). Here, either in solution or in the dilute phase of a phase-separated system, we did not detect evidence of large clusters (which, if present even at low abundance, would distort the FCS correlation functions at long time lags). We note that EPYC1-Rubisco is a two-component system with phase-separation driven by heterotypic interactions between the two

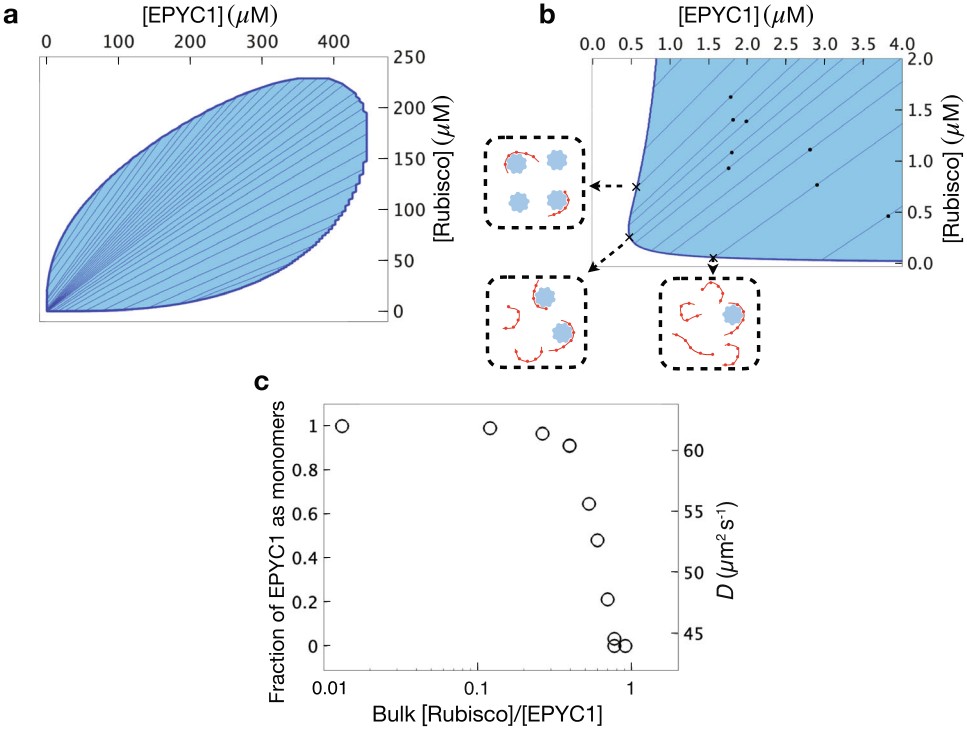

**Fig. 4 Predictions of a minimal model for the EPYC1-Rubisco system. a** The full modeled phase diagram is shown as a function of total concentrations of EPYC1 polymers and Rubisco holoenzymes. The two-phase region is shaded and each internal tie line connects the dense and dilute phases corresponding to all bulk concentrations along that line. **b** Zoomed-in version of the phase diagram showing the same range as the experiments in Figs. 1c and 3c. The boxes outside of the middle panel illustrate the contents of the dilute phase at the marked points on the dilute-phase boundary. **c** The fraction of EPYC1 in the dilute phase that are monomers, and the corresponding semi-empirical prediction for the EPYC1 diffusion coefficient as functions of the overall concentration ratio of Rubisco and EPYC1 for the black dots in **b** which were chosen to closely match the experimental concentrations in Fig. 3c.

molecules, while the examples tested in Kar et al. were all single-component systems. Whether this notable difference can account for the different observations requires further study.

An interesting feature of the EPYC1-Rubisco complexes (Figs. 2, 3) is that each complex seems to only contain one Rubisco molecule, even at high Rubisco/EPYC1 concentration ratios. In other words, EPYC1 does not seem to bridge multiple Rubiscos in the dilute phase. This can be understood as follows. First, localizing additional Rubiscos would cost translational entropy. Second, when one EPYC1 binding site interacts with one Rubisco binding site, it is more likely that other binding sites of the same EPYC1 will interact with the same Rubisco, instead of bridging to other Rubiscos. Because of this subunit cooperativity, additional Rubiscos would not necessarily lower the binding free energy of the EPYC1. While our results suggest that the dilute-phase complexes contain one Rubisco, the number of EPYC1s in each complex remains unknown. To resolve the number of EPYC1s per complex, techniques such as cryo-electron microscopy or single-molecule FRET could be employed in future studies.

We further show that a simple analytical model recapitulates all experimental findings. The model includes single EPYC1s and Rubiscos as well as EPYC1-Rubisco heterodimers in the dilute phase but, motivated by the experimental results, neglects higher-order complexes such as Rubisco with multiple EPYC1s. The model is also evaluated in mean-field which neglects correlations in the dense phase and includes a minimization over two limits, one where molecular heterodimers dominate and one where independent sticker pairs dominate. Including corrections to this mean-field model could provide additional insight into detailed bonding arrangements in the dense phase as studied for neuronal proteins[32]. Our experiments do not find ternary phase separation

as found in other contexts such as electrostatic models of intrinsically disordered proteins[33]. Future studies could include higher order complexes, explicit solvent, and corrections to the mean-field model to more fully characterize both the dense and dilute phases.

To our knowledge, this work is the first experimental investigation of dilute-phase organization in a biphasic system at equilibrium and our findings pave the way to further experimental and theoretical investigations of how complexes in the dilute phase could regulate condensate stability and dynamics. For example, recent advances in the field have demonstrated many elegant ways to experimentally measure tie-lines[32,34,35]. These measurements should also be possible for the EPYC1-Rubisco system studied here and would enable more detailed and quantitative testing of theoretical models and simulations[13,36,37].

This work could also have important implications to pyrenoid biology. At cell division, much of the pyrenoid disassembles then reassembles in daughter cells[11], presumably to facilitate splitting of the residual pyrenoid and/or to ensure each daughter has material to form a new pyrenoid. This disassembly implies a large increase of Rubisco in the dilute phase. It is not yet clear what mechanism underpins pyrenoid disassembly, but we envision two distinct scenarios: (i) posttranslational modifications of EPYC1 could generally weaken EPYC1-Rubisco interactions, so condensates become unstable, and complexes in the dilute phase also become *less* stable, or (ii) modifications of EPYC1 could favor dilute phase complexes, so that condensates fall apart because the competing dilute phase complexes become *more* stable. Future study is needed to distinguish between these two scenarios and more generally to investigate how post-translational modifications regulate dilute-phase complexes and the overall phase behavior of the pyrenoid.

## Methods

**Protein expression and purification**. Concentrations of protein samples were calculated from absorbance at 280 nm (488 nm for EPYC1-GFP) using extinction coefficient obtained from the ExPASy-ProtParam platform (https://web.expasy.org/protparam/). All proteins were buffer exchanged into 20 mM Tris-HCl, 50 mM NaCl buffer at pH 8.0 (BufferA) using a Micro BioSpin 30 (BioRad) chromatography column for EPYC1 or size-exclusion chromatography (Superdex 200 10/300 GL, Cytiva) for Rubisco and EPYC1-GFP.

The plasmids expressing EPYC1 and EPYC1-GFP are pHueCrEPYC1 and pHueCrEPYC1-GFP, respectively[18]. Both plasmids encode residues 46-317 of EPYC1, excluding transit peptide. GFP is enhanced GFP with monomerizing A206K mutation[38].

EPYC1 and EPYC1-GFP proteins were expressed in BL21 DE3 E.coli cells transformed with pHueEPYC1 and pHueEPYC1-GFP plasmid. Cells were grown in LB + antibiotics at 37 °C overnight, diluted 100-fold the next morning and incubated for 2.5 hours to reach OD600 between 0.4-0.8, then 0.4 mM IPTG was added and the cells were incubated for 3 hours at 37 °C to induce protein expression. Cells were harvested using centrifugation at 5000 RCF for 10 minutes and resuspended in high-salt lysis buffer [20 mM Tris-HCl, 500 mM NaCl, 10 mM Imidazole, 0.3 mg/ml Lysozyme, 3 mM phenylmethylsulfonyl fluoride (PMSF) with 50 units Benzonase (Sigma Aldrich), 2 mM MgCl₂]. Lysozyme improves cell lysis efficiency. PMSF is a protease inhibitor. Benzonase with adequate Mg²⁺ degrades and eliminates nucleic acid contamination, which induces EPYC1 aggregation. After sonication and centrifugation, the supernatant of the cell lysate was loaded onto Ni-NTA agarose resin (Qiagen) in Poly-prep chromatography columns (BioRad). The sample was washed with the same lysis buffer without Lysozyme and PMSF and eluted with high-salt 300 mM imidazole buffer. For long-term storage, EPYC1 was buffer exchanged into storage buffer (20 mM Tris-HCl pH 8.0, 50 mM NaCl buffer with 5% glycerol) using a size-exclusion column (Superdex 200 10/300 GL, Cytiva) before flash-frozen in liquid nitrogen and stored at −80 °C.

Wildtype and mutant Rubisco were extracted from *Chlamydomonas reinhardtii* cells (wildtype: CC4533 cw15, mt-; mutant: M87D/V94D Rubisco small subunits point mutant in T60-3 Δrbcs background[16]) grown in TAP medium at high CO₂. Cells were harvested in extraction buffer (10 mM MgCl₂, 50 mM Bicine pH 8.0, 10 mM NaHCO₃, 1 mM Dithiothreitol (DTT)) with protease inhibitor cocktail (Roche), flash frozen in liquid nitrogen, and stored at −80 °C. For Rubisco purification, cells were lysed using a CryoMill (Retsch) at 15 Hz for 15 minutes. The soluble fraction was loaded on a 10–30% sucrose gradient and ultracentrifuged in a SW41 Ti rotor at a speed of 35,000 rpm for 20 hours at 4 °C. The fractions containing Rubisco were obtained and further purified by monoQ anion exchange chromatography (Cytiva). The protein was buffer exchanged using size-exclusion chromatography prior to being flash-frozen for long-term storage.

**Rubisco labeling**. To determine the diffusion coefficients of Rubisco, and test which surface condition works the best for the proteins in this study, we labeled Rubisco with amine-reactive Alexa 488 (Invitrogen, A10235) and removed free dyes using Micro BioSpin 30 chromatography column. The degree of labelling is ~2, which is determined by the ratio of dye concentration (from 488 nm absorption) and protein concentration (from 280 nm absorption).

**Droplet turbidity assay**. 30 µl EPYC1 and 30 µl Rubisco proteins were mixed in the cuvette for measurement in the Cary 60 UV-Vis spectrophotometer (Agilent). After 10 minutes, the extinction at 340 nm was measured to determine if there was phase separation[39]. The presence of protein condensates leads to an increase of extinction at 340 nm. If extinction at 340 nm was above 0.1 after 10 minutes of mixing, the sample was considered phase separated.

**Fluorescence Correlation Spectroscopy (FCS)**. FCS was conducted on a custom-built setup similar to previously described[40]. Specifically, a 488 nm laser (Coherent Sapphire, 20 mW) was focused onto the sample solution using an oil immersion objective (1.49NA, Olympus). Typical laser power used was ~90 µW at the back aperture of the objective, which gives a peak power density of ~10 kW/cm². Emitted fluorescent photons were collected by the same objective, focused on a 50 µm pinhole, spectrally filtered (HQ525/50, Chroma), and refocused onto an avalanche photodiode detector (Excelitas SPCM-AQRH-24-TR).

The glass coverslip (Thorlabs #1.5H, CG15CH) was cleaned with Piranha solution (3:1 mixture of sulfuric acid and hydrogen peroxide) followed by air plasma treatment (Harrick Plasma PDC-32G, 600 mTorr air plasma, 5 min) prior to immersion in a 1 mg/ml PEI-PEG (Sigma-Aldrich 900743 or NanoSoftPolymers 12107-25K-5000-20-100 mg) in PBS (Invitrogen) buffer. The PEI-PEG coating is essential to minimize protein sticking for both EPYC1 and Rubisco. After immersion for 10 min, the coverslip was rinsed and blown dry with ultrapure N₂ and attached to a hybridization chamber (Grace Bio-labs SecureSeal GBL621505). Typically ~15 µl samples were applied to the chamber for measurements. FCS measurements were performed ~5 µm above the coverslip surface to minimize signal from the coverslip surface.

FCS data were typically collected for 1 min and repeated 3-5 times under identical conditions. For experiments that mixed EPYC1-GFP and EPYC1, the

concentration of the labeled species (EPYC1-GFP) was fixed at 20 nM to ensure consistent FCS data quality across experimental conditions.

The FCS autocorrelation curves were computed using custom-made Matlab programs[40]. A simple model assuming one diffusive species without triplet or photophysical dynamics of the label was used to fit the autocorrelation curves:

$$G(\tau) = \frac{G_0}{\left(1 + \frac{\tau}{\tau_D}\right)\left(1 + \left(\frac{w_{xy}}{w_z}\right)^2 \frac{\tau}{\tau_D}\right)^{1/2}} + c_0$$

In this equation, $G_0$, $\tau_D$, and $c_0$ are the fitting parameters. $w_{xy}$ and $w_z$ are the lateral and axial $1/e^2$ radius of the detection volume. $G_0$ is the autocorrelation amplitude at zero delay ($\tau = 0$), which was used to normalize the autocorrelation curves in Fig. 2 and Supplementary Fig. 3. $\tau_D$ is the characteristic diffusion time and is related to the diffusion coefficient by $D = w_{xy}^2/4\tau_D$. The focal volume of the FCS setup was measured to be 0.58 fL using Atto488 samples in a dilution series with known concentrations[41] (Supplementary Fig. 2a).

**Wide-field epifluorescence microscopy**. To directly image the fluorescent droplets, a lens (f = 300 mm) was inserted into the custom-built FCS microscope excitation path to focus the laser to the back focal plane of the objective lens, generating a ~30 µm diameter illumination field[42]. Images were captured on a sCMOS camera (FLIR CM3-U3-31S4M-CS) using a typical integration time of 10–50 ms. To conduct subsequent FCS experiments at desired locations in the dilute phase, the sample was translated laterally using a piezo stage (PI P-563), and the pre-focusing lens was removed. All microscope control and synchronization tasks were accomplished using custom-written software in LabView.

**Statistics and reproducibility**. All measurements reported here were repeated for at least 3 times. Each FCS measurement was conducted at 20 nM concentration of the fluorescently labeled species for 60 seconds to ensure a statistically robust number of molecules were sampled (>60,000) during the measurement. Another important factor that determines reproducibility of this work is the nature and quality of the surface preparation protocol used. As detailed in the main text, a new surface coating protocol was developed and rigorously tested to ensure data reproducibility.

**Analytical model**. We now briefly describe the full dimer-gel model[13] and then describe the approximate form used in this work. The model free-energy density is composed of three terms:

$$F = F_{ni} + F_{ex} + F_s. \quad (1)$$

The first term in Eq. 1 is the non-interacting, purely entropic, contribution to the free energy,

$$\frac{F_{ni}}{k_B T} = \frac{c_R}{L_R} \log \frac{c_R}{eL_R} + \frac{c_E}{L_E} \log \frac{c_E}{eL_E}, \quad (2)$$

where $L_R = 8$ and $L_E = 5$ are the total number of stickers on a Rubisco and on an EPYC1, respectively, $c_R$ and $c_E$ are their total sticker concentrations. The analogous total polymer concentrations are $\rho_R = \frac{c_R}{L_R}$ and $\rho_E = \frac{c_E}{L_E}$ (written as [Rubisco] and [EPYC1] in the main text). We employed the following expression for the excluded-volume interaction (termed "nonspecific interactions" in Zhang et al.[13]):

$$\frac{F_{ex}}{k_B T} = v_R c_R^2 + v_E c_E^2 + v_{ER} c_E c_R, \quad (3)$$

which for simplicity considers all stickers as hard-sphere monomers. The effective volume coefficients are obtained from a virial expansion[43,44] such that $v_R$ for Rubisco-Rubisco excluded volume is taken to be 4 times one-eighth of the volume of a spherical Rubisco with a diameter of $d_R = 10$ nm, yielding $v_R = 4 \frac{4}{3} \frac{\pi}{8} \left(\frac{d_R}{2}\right)^3 = 261.80 \text{nm}^3$. An EPYC1 sticker includes the length of the region that binds Rubisco[16] plus the length of a linker, for a total of ~60 amino acids each about 3-4 Å long[45]. This gives an approximate radius of gyration of a model EPYC1 sticker to be $R_g = 1$ nm, yielding an EPYC1-EPYC1 effective volume coefficient of $v_E = 4 \frac{4}{3} \pi R_g^3 = 16.76 \text{nm}^3$. For the excluded volume coefficient between Rubisco and EPYC1 stickers we express $v_{ER}$ in terms of the volume of a sphere whose radius is given by the average of the effective radii of the two stickers, yielding $v_{ER} = 8 \frac{4}{3} \pi \left(\frac{d_R}{8} + \frac{R_g}{2}\right)^3 = 179.60 \text{nm}^3$. All the volumes are written in terms of molarity by converting to liters and multiplying by Avogadro's number. The third term in Eq. 1 captures the specific interactions between EPYC1 stickers and Rubisco stickers:

$$\frac{F_s}{k_B T} = -\frac{1}{V} \ln Z_s, \quad (4)$$

where the partition function is

$$Z_s = P(N_{dR}, N_{dE}, N_b) W(N_{dR}, N_{dE}, N_b) \exp\left(\frac{N_{dR} \varepsilon_d}{L_R} + N_b \varepsilon_b\right) \quad (5)$$

where $N_{dR}$ is the number of Rubisco stickers in dimers, $N_{dE}$ is the number of EPYC1 stickers in dimers, $\varepsilon_b$ is the effective binding energy for sticker pairs, $\varepsilon_d$ is the effective binding energy for dimers, and $N_b$ is the number of additional sticker pairs. The probability distribution function, $P(N_{dR}, N_{dE}, N_b)$, accounts for the many ways that polymers can for dimers and stickers can form independent bonds,

$$P(N_{dR}, N_{dE}, N_b) = \binom{N_R/L_R}{N_{dR}/L_R}\binom{N_E/L_E}{N_{dE}/L_E}(N_{dR}/L_R)! \\ \binom{N_R - N_{dR}}{N_b}\binom{N_E - N_{dE}}{N_b}N_b! \quad (6)$$

Here, the number of total stickers for Rubisco is $N_R$ and the total stickers for EPYC1 is $N_E$. Second, $W(N_{d1}, N_{d2}, N_b)$ is the probability distribution function that polymers are close enough in space to form dimers and stickers are close enough in space to form independent bonds,

$$W(N_{d1}, N_{d2}, N_b) = \left(\frac{\nu_d}{V}\right)^{\frac{N_{dR}}{L_R}}\left(\frac{\nu_b}{V}\right)^{N_b}, \quad (7)$$

where $\nu_d$ is the effective interaction volume for dimers and $\nu_b$ is the effective interaction volume for independent sticker pairs. Combining Eqs. 4–7 and using Stirling's approximation the free energy in Eq. 3 yields

$$\frac{F_s}{k_B T} = -\frac{c_R}{L_R}\ln c_R + (1 - L_R)\frac{c_R - c_{dR}}{L_R}\ln(c_R - c_{dR}) \\ + (c_R - c_{dR} - c_b)\ln(c_R - c_{dR} - c_b) \\ - \frac{c_E}{L_E}\ln c_E + (1 - L_E)\frac{c_E - c_{dE}}{L_E}\ln(c_E - c_{dE}) \\ + (c_E - c_{dE} - c_b)\ln(c_E - c_{dE} - c_b) + \frac{c_{dR}}{L_R}\ln(ec_{dE}L_R K_d) \\ + c_b\ln(ec_b K_b), \quad (8)$$

where the dissociation constants for dimers and independent sticker pairs is $K_d \equiv e^{\varepsilon_d}/\nu_d$ and $K_b \equiv e^{\varepsilon_b}/\nu_b$. The free energy for specific bonds will be minimized over $c_b$, $c_{dR}$, and $c_{dE}$ in the thermodynamic limit giving the following constraint:

$$c_{dE}L_R K_d = (c_R - c_{dR} - c_b)^{L_R}(c_E - c_{dE} - c_b)^{L_E}(c_R - c_{dR})^{1-L_R}(c_E - c_{dE})^{1-L_E} \\ c_b K_b = (c_R - c_{dR} - c_b)(c_E - c_{dE} - c_b).$$

Inserting the constraint into Eq. 8 gives the specific free energy

$$\frac{F_s}{k_B T} = \frac{c_R}{L_R}\ln\frac{(c_R - c_{dR})}{c_R} + \frac{c_E}{L_E}\ln\frac{(c_E - c_{dE})}{c_E} + c_R\ln\frac{(c_R - c_{dR} - c_b)}{(c_R - c_{dR})} \\ + c_E\ln\frac{(c_E - c_{dE} - c_b)}{(c_E - c_{dE})} + \frac{c_{dR}}{L_R} \quad (9)$$

The simplified model used in this work employs the free energy given by Eq. 1, but simplifies Eq. 9 to

$$\frac{F_s}{k_B T} = \min(F_{dim}, F_{ind}).$$

For the specific free energy, $F_s$, we use whichever is the minimum of the dimer-bond free energy, $F_{dim}$, or the independent-sticker-bond free energy, $F_{ind}$. In Ref. 13, this simplified form was motivated by a similar model where Eq. 9 was found to be saturated by one of the two free energies described above. To gain intuition for the EPYC1-Rubisco system, we can compute both free energies at equal stoichiometry in both the dilute and dense phases at, respectively, approximately $\rho_{dilute} = 0.5\,\mu M$ and $\rho_{dense} = 150\,\mu M$. In the dense phase, we find that the independent-sticker-bond free energy is dominant and the probability of finding a dimer instead of independent stickers is negligible at $\sim e^{-6}$ (we obtain this estimate by multiplying the difference in free-energy densities by the volume a dimer would occupy in the dense phase, $1/\rho_{dense}$). Similarly, in the dilute phase, the model probability of finding a dimer dissociated into stickers is also small, $\sim e^{-2}$ and can reasonably be neglected (in this case the free-energy difference was multiplied by $1/\rho_{dilute}$). Hence, allowing for the coexistence of dimers and independent sticker bonds in a single phase will not substantially affect the phase diagram, so for simplicity and clarity we have chosen to keep only the dominant contribution in each phase. In the simplified model, the free-energy density for a homogeneous solution of two interacting polymer species is therefore given by the following:

$$F = F_{ni} + F_{ex} + \min(F_{dim}, F_{ind}).$$

The two specific free-energy contributions to the specific interactions in the third term on the RHS are given as follows:

$$\frac{F_{dim}}{k_B T} = \rho_d \ln K_d + \rho_d \ln\frac{\rho_d}{e} + (\rho_R - \rho_d)\ln\frac{(\rho_R - \rho_d)}{e} + (\rho_E - \rho_d)\ln\frac{(\rho_E - \rho_d)}{e} \\ - \rho_E \ln\frac{\rho_E}{e} - \rho_R \ln\frac{\rho_R}{e}, \quad (10)$$

and

$$\frac{F_{ind}}{k_B T} = c_b \ln K_b + c_b \ln\frac{c_b}{e} + (c_R - c_b)\ln\frac{(c_R - c_b)}{e} + (c_E - c_d)\ln\frac{(c_E - c_d)}{e} \\ - c_E \ln\frac{c_E}{e} - c_R \ln\frac{c_R}{e}, \quad (11)$$

where the concentration of dimers in polymeric units, $\rho_d$, is

$$\rho_d = \frac{1}{2}\left[\rho_R + \rho_E + K_d - \sqrt{(\rho_R + \rho_E + K_d)^2 - 4\rho_R \rho_E}\right]$$

and the concentration of bonded stickers in sticker units, $c_b$, is

$$c_b = \frac{1}{2}\left[c_R + c_E + K_b - \sqrt{(c_R + c_E + K_b)^2 - 4c_R c_E}\right]$$

The specific interactions have as inputs the dissociation constant between an EPYC1 polymer and a Rubisco holoenzyme, $K_d$, and the sticker-sticker dissociation constant between individual EPYC1 and Rubisco stickers, $K_b$. We use the experimental value for $K_d$, and use $K_b$ as the only free fitting parameter.

The dilute phase boundary as shown in Fig. 4a, b yields predictions for the occupants of the dilute phase. Given the strong binding between EPYC1 and Rubisco almost all possible dimer pairs will form, and so it is simple to infer the amount of free EPYC1 in the dilute phase. Specifically, the fraction of EPYC1 as monomers is given by the excess relative EPYC1 concentration to Rubisco concentration divided by the total EPYC1 concentration, all evaluated at the dilute-phase boundary. From this detailed information, predictions for the EPYC1 diffusion constant can be made by assuming that the diffusion constant is the average over the EPYC1 dilute-phase complexes. Specifically, the diffusion constant is the fraction of EPYC1 as monomers multiplied by the EPYC1 monomer diffusion constant plus the fraction of EPYC1 as dimers multiplied by the dimer diffusion constant. From Fig. 3d, the diffusion of EPYC1 as monomers is ~62 $\mu m^2/s$ and of EPYC1 as EPYC1-Rubisco heterodimers is ~42 $\mu m^2/s$.

**Reporting summary.** Further information on research design is available in the Nature Portfolio Reporting Summary linked to this article.

## Data availability

Data supporting the findings of this manuscript are available in the Supplementary Data for data underlying Figs. 2, 3, and from the corresponding author upon reasonable request.

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

## Acknowledgements

We thank Cliff Brangwynne, Alexei Korennykh, Ming-Tzo Wei, and members of the Jonikas, Wang, and Wingreen laboratories for insightful discussions. This work was supported by grants from the National Institutes of Health (R01GM140032), the Howard Hughes Medical Institute and Simons Foundation (55108535), the National Science Foundation (MCB-1935444), and the National Science Foundation through the Center for the Physics of Biological Function (PHY-1734030). H.W. and Q.W. acknowledge support from the Lewis-Sigler Fellowship of Princeton University. G.H. was supported by a China Scholarship Council scholarship. T.G. was supported by the Schmidt Science Fellowship. M.C.J. is a Howard Hughes Medical Institute Investigator.

## Author contributions

G.H., M.C.J., N.S.W., and Q.W. designed the study. G.H. performed experiments. T.G., J.Z., and N.S.W. developed the theoretical model. H.W. constructed the FCS apparatus. All authors contributed to interpreting results and preparing the manuscript.

## Funding

## Competing interests

The authors declare no competing interests.

## Additional information

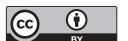

