## [Peer Review File · Communications Biology]

Reviewers' comments:

Reviewer #1 (Remarks to the Author):

Liquid-liquid phase separation (LLPS) is thought to drive the formation of various membraneless biomolecular condensates *in vivo*. Most studies on LLPS have focused on characterizing the properties of the condensates in the dense phase. However, recent computer simulations suggested that small complexes dominate the dilute phase and are important in regulating phase separation. In this paper, to visualize such complexes experimentally He et al. investigated *in vitro* the dilute phase of a two-component LLPS system containing Rubisco and EPYC1, which are the main components in the pyrenoid of algal chloroplasts. Using fluorescence correlation spectroscopy (FCS) as their main method, they found Rubisco-EPYC1 complexes in the dilute phase. By analyzing the EPYC1 diffusion coefficient, they extracted the dissociation constant and stoichiometry of the complexes from the FCS data, and found that the majority of the complexes in the dilute phase contained one Rubisco molecule. They also developed an analytical free-energy model to better understand the organization of the dilute phase.

This study shows that FCS is a powerful technique for the analysis of the dilute phase of biomolecular condensates *in vitro*. However, the question how the protein complexes in the dilute phase regulate condensate formation is still open.

Major points:

1. I assume that the diffusion rates shown in Fig. 2e were obtained from at least 3 independent experiments for each Rubisco concentration. The error bars for these measurements should be shown rather than the horizontal bars for the estimated uncertainty of the Rubisco concentration.

2. The authors should consider also using the EPYC1 mutant (He. S et al; <https://doi.org/10.1038/s41477-020-00811-y>) which was shown to have a weaker binding affinity for Rubisco. The dissociation constant of this mutant is expected to be higher, and could be used to validate the reliability of the FCS analysis in this study. Moreover, the comparison of the interaction of the mutant EPYC1 and wild-type EPYC1 with Rubisco may provide insights to understand how the changes to binding affinity regulate condensate formation – as mentioned by the authors in the discussion of the two distinct scenarios (page 20).

3. When the authors analysed the dilute phase of a phase-separated system, EPYC1-GFP concentration was fixed at 20 nM and EPYC1 concentration varied from 1 – 4 μ M. This results in the ratio of EPYC1-GFP/EPYC1 to change from 0.02 - 0.005. Does the measurement time remain the same? How does it affect the number of counts/events?

4. Page 18, sentence "... where EPYC1 concentrations are higher than Rubisco concentrations all Rubiscos will form dimers, leaving an excess of EPYC1s." I suppose the authors mean Rubisco forms heterodimers?

5. Authors may want to compare their findings with the recent report on cluster formation in the subsaturated solutions of phase-separating RNA-binding proteins (Mrityunjoy Kar et al PNAS; <https://doi.org/10.1073/pnas.2202222119>).

Minor points:

1. The full name of Rubisco should be written out at least once in the manuscript.

2. SDS-PAGE of the purified proteins EPYC1, EPYC1-GFP and Rubisco used in this study should be shown as Supplementary Fig.

3. Page 8, last line: "Fig. 2c inset" should be Fig. 2d.

4. Page 9: "To quantify..... (Fig. 2d)." should be Fig. 2e.
5. Page 12: Sentence "We mixed EPYC1-GFP (fixed at 20 nM), unlabeled EPYC1 (from 1 to 4 μ M), and Rubisco (ranging from 0.05 to 2.5 μ M)", however, as shown in Fig. 3c the highest concentration of Rubisco used was only 2 μ M.
6. Model in Fig. 3a should also show free EPYC1-GFP in the dilute phase.
7. Is it possible to change the numbers of the x-axis in Fig. 3d to 0.01, 0.1 and 1? The ticks for the log scale are hardly visible. This applies to all figures.
8. Legend Fig. 3d, please state the number or range of repeated experiments for the error bars.
9. Materials and Methods, page 21: Please check the plasmid for expressing EPYC1-GFP (pHueFLAGCrEPYC1??)
10. Legend of supplementary Figure 3: There are no panels d and e in the figure. The x-axis numbering for Rubisco concentration in Fig. S3b is incorrect.

Reviewer #2 (Remarks to the Author):

The manuscript by He et. al. describes the role of dilute phase protein complexes in the regulation of protein condensates. By applying the fluorescence correlation spectroscopy (FCS) with a newly developed analytical model, the authors found that the Rubisco and EPYC1 form complexes in the dilute phase no matter if the proteins form condensations or not. Moreover, by analyzing the diffusion coefficients, the authors characterize the composition of the complexes in the dilute phase. This work provides an interesting investigation for understanding how proteins organize in the dilute phase upon co-condensation. To strengthen the work, I have several comments listed below.

1. The author applied FCS to determine the binding affinity of Rubisco and EPYC1. However, this result shows \pm 68% confidence interval, which is relatively low. So, how is the accuracy of the analytical model which is based on the dissociation constant of EYPC1 and Rubisco measured by FCS data.
2. Because of the large difference in molecular weight (MW) of Rubisco (550 kDa) and EPYC1 (35 kDa), the author can deduce the complexes' composition from the diffusion coefficients. How about the complexes formed by two different proteins with close MW? Can the FCS method be applied to precisely characterize the composition of the complex?
3. As the dilute phase in vitro is homogeneous, the composition of the dilute phase can be determined as shown in this study. Whether this approach can be applied in characterizing the dilute phase in the cellular microenvironment where condensates formed.

Reviewer #3 (Remarks to the Author):

Report on Comm Biol Ms by He et al. & Wang, Ms. No.: COMMSBIO-22-2777-T

The work reported in this manuscript on molecular association in the dilute phase in equilibrium with a condensed biomolecular phase is timely and interesting. Using a Rubisco-EPYC1 model system, the experimental results shed new light on the property of such dilute phases, finding Rubisco-EPYC1 dimers in the dilute phase that may compete with condensation. These measurements are highly

valuable as they help advance knowledge about the role of dilute-phase properties in biomolecular condensation in general. However, there are substantial room for improvement in the experimental set up and theoretical modeling--at least in presentation of the results if a more complete study is deemed to be beyond the scope of this manuscript. The issues are related to a lack of measurement of condensed-phase concentration, and therefore a lack of experimental tie-lines (which would be needed for a direct comparison with the theoretical model result in Fig.4a,b). The theoretical model as it stands does not rationalize dimer formation and phase separation using a general physical interaction, but rather accounting for them in two different terms. While some of these shortcomings may be dealt with in future work, their limitations and possible pathways forward should be discussed in the context of closely related works in the literature. More specifically, the following issues should be addressed in revising this manuscript:

1. Is there a rigorous way to translate the experimentally measured diffusion coefficients into EPYC1-Rubisco dimer concentration in the dilute phase? What is the stoichiometry of dilute-phase association? These points are not clear in the manuscript.
2. If there is such a correspondence, it will be useful to also show a version of Fig.3c in terms of dimer concentration, not just diffusion coefficients.
3. Why are there more data points in Fig.3c (for $[\text{Rubisco}] > 1\mu\text{M}$) than in Fig.1d? If there are phase-separated states for $[\text{Rubisco}] > 1\mu\text{M}$, they should be plotted in Fig.1d as well.
4. Because there is no measurement of concentrations of Rubisco and EPYC1 in co-existing condensed phases, there are no experimental tie-lines to compare with the theoretically predicted tie-lines in Fig.4. Condensed-phase protein concentrations have been measured experimentally [see, e.g., Brady et al., Proc Natl Acad Sci USA 114:E8194-E8203 (2017)], and more recently for a two-biomolecule SynGAP-PSD95 model for postsynaptic densities (PSD) [Lin et al., Biophys J 121:157-171 (2022)]. Can these published methods be applied to measure co-existing condensed-phase densities for the authors' EPYC1-Rubisco system?
5. The role of dilute-phase stoichiometric binding in the assembly of biomolecular condensates has been addressed recently using the SynGAP-PSD95 PSD model mentioned above (Biophys J 2022). This work is highly relevant and should be discussed and compared with work reported in the present manuscript. In particular, the PSD model indicates, based on comparison between experimental and theoretical tie-lines (in several different theoretical scenarios), that auxiliary interactions in addition to dilute-phase stoichiometric binding are needed to account for the assembly of the condensed phase.
6. Indeed, the tie-line pattern exhibited by the current model in Fig.4 corresponds roughly to those shown in Fig.5 in the recent Biophys J 2022 study for the presence of auxiliary interactions – that's consistent with the current theoretical set-up that, quite artificially (see below), assumes that the physical forces driving dimer formation and forces driving condensation as independent.
7. Possible tie-line patterns (diverging, converging, positive slopes, negative slopes etc.) for two-solute system has been explored in the biomolecular condensates by Lin et al., New J Phys 19:115003 (2017), including how different tie-line patterns for co-mixing and de-mixing are governed by matching/mismatching of charge patterns of a pair of polyampholyte species (Fig.3 of this reference), different Flory-Huggins interaction parameters (Figs.5 and 6 of this reference), and the possibility of ternary phase separation (Fig.7a of this reference). Is ternary phase separation possible for the current study's EPYC1-Rubisco system? In any case, to put the present work in the context of recent progress in the exploration of two-biomolecular-species phase separation, the scenarios in this earlier New J Phys 2017 work should be discussed.
8. The theoretical model introduced in the current manuscript uses a free energy function with the $\min(F_{\text{dim}}, F_{\text{ind}})$ term (Eq.1). While this term (together with the other two terms in the free energy

function) is useful for providing a qualitative account for the experimental trend (Fig.4a,b), the usage of two different free energies, F_{dim} and F_{ind} , to describe the physical forces responsible for dimer formation on one hand and the physical forces ("sticker-sticker" interactions) for condensation on the other hand means that the experimental data are NOT accounted for by general physical interactions governing BOTH dimer formation AND condensation (that's how it should work in the real physical world). By doing so, the modeling effort might have missed the opportunity to suggest a possible presence of Rubisco-Rubisco and EPYC1-EPYC1 in the condensed phase (cf. Biophys J 2022 study mentioned above).

9. Physically, both dimer formation and the "sticker-sticker" contacts must arise from the same general physical interaction. Here, in the authors' model (Eq.1), the "min" prescription means that there is an abrupt transition from the system being described by F_{dim} to being described by F_{ind} , just when $F_{dim} - F_{ind}$ turns from a very small positive number to a very small negative number, and vice versa. That means the physical forces responsible for dimer formation play no role in condensation in the theoretical model. That's not reasonable physically. Mathematically, this theoretical set-up might have created an artificial "all-or-none" feature in the model that helps with fitting experimental trend but does not accurately capture the physical interactions involved. These limitations of the theoretical model should be addressed and possible improvements discussed.

Reviewer #1 (Remarks to the Author):

Liquid-liquid phase separation (LLPS) is thought to drive the formation of various membraneless biomolecular condensates in vivo. Most studies on LLPS have focused on characterizing the properties of the condensates in the dense phase. However, recent computer simulations suggested that small complexes dominate the dilute phase and are important in regulating phase separation. In this paper, to visualize such complexes experimentally He et al. investigated in vitro the dilute phase of a two-component LLPS system containing Rubisco and EPYC1, which are the main components in the pyrenoid of algal chloroplasts. Using fluorescence correlation spectroscopy (FCS) as their main method, they found Rubisco-EPYC1 complexes in the dilute phase. By analyzing the EPYC1 diffusion coefficient, they extracted the dissociation constant and stoichiometry of the complexes from the FCS data, and found that the majority of the complexes in the dilute phase contained one Rubisco molecule. They also developed an analytical free-energy model to better understand the organization of the dilute phase.

This study shows that FCS is a powerful technique for the analysis of the dilute phase of biomolecular condensates in vitro. However, the question how the protein complexes in the dilute phase regulate condensate formation is still open.

We thank the reviewer for the favorable evaluation of this work. We agree with the reviewer that the question of how complexes in the dilute phase influence condensate formation is still open, but we believe this work has made an important first step towards this goal.

Major points:

1. I assume that the diffusion rates shown in Fig. 2e were obtained from at least 3 independent experiments for each Rubisco concentration. The error bars for these measurements should be shown rather than the horizontal bars for the estimated uncertainty of the Rubisco concentration.

Thank you for this suggestion. We have modified Fig. 2e to display error bars calculated from 3 independent experiments. We note that the horizontal error bars are important for quantitative evaluation of our data, because they represent our best efforts to account for the discrepancy (due to nonspecific surface interactions) between actual and nominal Rubisco concentrations.

Fig. 2e has been updated (included below for the reviewer's convenience) and the caption to Fig. 2e now reads:

“e Diffusion coefficient (D) of EPYC1-GFP inferred from FCS as a function of Rubisco concentration. Vertical error bars are standard deviations of repeated experiments ($n=3$).”

2. The authors should consider also using the EPYC1 mutant (He. S et al; <https://doi.org/10.1038/s41477-020-00811-y>) which was shown to have a weaker binding affinity for Rubisco. The dissociation constant of this mutant is expected to be higher, and could be used to validate the reliability of the FCS analysis in this study. Moreover, the comparison of the interaction of the mutant EPYC1 and wild-type EPYC1 with Rubisco may provide insights to understand how the changes to binding affinity regulate condensate formation – as mentioned by the authors in the discussion of the two distinct scenarios (page 20).

This is an excellent suggestion. We agree that a detailed, systematic study of the link between EPYC1-Rubisco affinity and phase separation behavior would provide new insights. These studies would require GFP-fused mutant EPYC1 constructs, which are not yet available and also might be better suited for a separate study due to the potential complexity involved (e.g., EPYC1 contains 5 Rubisco-binding regions, each of which contains candidate sites for mutation).

On the other hand, we were able to do another set of experiments which are along the line of the reviewer’s suggestion. In the same paper that the reviewer quoted (He et al. Nat. Plants), it was shown that mutation of the Rubisco small subunit (in particular the M87D/V94D double mutant) almost completely disrupts EPYC1-Rubisco interaction and pyrenoid formation *in vivo*. We thus conducted FCS experiments using the M87D/V94D mutant Rubisco in place of the wild-type Rubisco. We found that in contrast to the wild-type Rubisco, which slows down EPYC1 diffusion due to binding, the mutant Rubisco has minimal effect on EPYC1’s diffusion. This observation is fully consistent with our previous work and at the same time validates the FCS analysis.

We have included this new data as Supplementary Figure 5, also shown below.

Supplementary Figure 5 | EPYC1 does not bind to a Rubisco mutant. **a** Cartoon showing the experimental setup: 10nM EPYC1-GFP is mixed with 13nM WT Rubisco or 13nM mutant Rubisco. The mutant Rubisco is the M87D/V94D Rubisco small subunit from He et al. 2020. **b** Diffusion rates of EPYC1-GFP alone, with 13nM WT Rubisco, or 13nM mutant Rubisco. The error bars are standard deviations of repeated experiments ($n=5$).

We added the reference to this new control experiment in the main text.

“Further, a Rubisco mutant, previously shown to almost completely disrupt EPYC1-Rubisco interaction and pyrenoid formation *in vivo*¹⁶, does not slow down the diffusion of EPYC1-GFP (Supplementary Fig. 5).”

We have also added in the online Methods section, “Protein expression and purification”, the source of the M87D/V94D mutant Rubisco:

“Wildtype and mutant Rubisco were extracted from *Chlamydomonas reinhardtii* cells (wildtype: CC4533 cw15, mt-; mutant: M87D/V94D Rubisco small subunit point mutant in T60-3 $\Delta rbcS$ background¹⁶) grown in TAP medium at high CO₂.”

3. When the authors analysed the dilute phase of a phase-separated system, EPYC1-GFP concentration was fixed at 20 nM and EPYC1 concentration varied from 1 – 4 μM . This results in the ratio of EPYC1-GFP/EPYC1 to change from 0.02 - 0.005. Does the measurement time remain the same? How does it affect the number of counts/events?

Yes, the reviewer is correct that the ratio of EPYC1-GFP/EPYC changes from 0.02 to 0.005 in these experiments. We note that FCS data quality is determined by the concentration of the fluorescent species

(here fixed to be 20 nM in all experiments). The measurement time is thus kept the same (1 minute) to ensure similar data quality across conditions.

We have added the following section in the online Method section, “Fluorescence Correlation Spectroscopy (FCS),” to clarify this point.

FCS data were typically collected for 1 min and repeated 3-5 times under identical conditions. For experiments that mixed EPYC1-GFP and EPYC1, the concentration of the labeled species (EPYC1-GFP) was fixed at 20nM to ensure consistent FCS data quality across experimental conditions.

4. Page 18, sentence “... where EPYC1 concentrations are higher than Rubisco concentrations all Rubiscos will form dimers, leaving an excess of EPYC1s.” I suppose the authors mean Rubisco forms heterodimers?

Yes, we have changed “dimers” to “heterodimers”.

5. Authors may want to compare their findings with the recent report on cluster formation in the subsaturated solutions of phase-separating RNA-binding proteins (Mrityunjoy Kar et al PNAS; <https://doi.org/10.1073/pnas.2202222119>).

We thank the reviewer for this suggestion. We agree and have added a comparison to this paper in the Discussion section.

“Kar et al.³¹ reported the existence of heterogeneous molecular clusters larger than ~100 nm in subsaturated solutions of FUS (and of other RNA-binding proteins with disordered domains). Here, either in solution or in the dilute phase of a phase-separated system, we did not detect evidence of large clusters (which, if present even at low abundance, would distort the FCS correlation functions at long time lags). We note that EPYC1-Rubisco is a two-component system with phase-separation driven by heterotypic interactions between the two molecules, while the examples tested in Kar et al. were all single-component systems. Whether this notable difference can account for the different observations requires further study.”

Minor points:

1. The full name of Rubisco should be written out at least once in the manuscript.

The sentence that first introduces Rubisco now reads:

“Prior studies showed that the pyrenoid is mainly composed of the CO₂-fixing enzyme, Ribulose-1,5-bisphosphate carboxylase/oxygenase (Rubisco),...”

2. SDS-PAGE of the purified proteins EPYC1, EPYC1-GFP and Rubisco used in this study should be shown as Supplementary Fig.

We thank the reviewer for raising this point. We have added the SDS-PAGE gel showing EPYC1, EPYC1-GFP and Rubisco in Supplementary Figure 1 (panel a).

a) Purified proteins on an SDS-PAGE gel. RBCL: Rubisco large subunit. RBCS: Rubisco small subunit. M: marker.

3. Page 8, last line: “Fig. 2c inset” should be Fig. 2d.

This is now fixed.

4. Page 9: “To quantify..... (Fig. 2d).” should be Fig. 2e.

This is now fixed.

5. Page 12: Sentence “We mixed EPYC1-GFP (fixed at 20 nM), unlabeled EPYC1 (from 1 to 4 μ M), and Rubisco (ranging from 0.05 to 2.5 μ M)”, however, as shown in Fig. 3c the highest concentration of Rubisco used was only 2 μ M.

We thank the reviewer for pointing out this inconsistency. We have modified the text to read:

“unlabeled EPYC1 (from 1 to 4 μM), and Rubisco (ranging from 0.05 to 2 μM)”

6. Model in Fig. 3a should also show free EPYC1-GFP in the dilute phase.

We have modified Fig. 3a to show EPYC1-GFP in the dilute phase.

7. Is it possible to change the numbers of the x-axis in Fig. 3d to 0.01, 0.1 and 1? The ticks for the log scale are hardly visible. This applies to all figures.

We have modified the x-axis labels of Fig. 3d according to the reviewer’s suggestions. We also changed the x-axis labels of Fig. 2c to increase readability. The major ticks now correspond to 0.01, 0.1, 1, 10, 100 milliseconds (ms). See the updated Fig. 2c below. We have also modified the axis ticks of Fig. 3d and 4c to increase visibility.

8. Legend Fig. 3d, please state the number or range of repeated experiments for the error bars.

We have added detailed description of the number of repeats in Fig. 3d for the error bar calculations. The figure caption of Fig. 3c and 3d now reads:

“**c** Diffusion coefficients (D) of EPYC1-GFP in the dilute phase in the presence of condensates obtained by FCS. Obtained D values are color coded as shades of blue (lighter blue represents higher D) and plotted on the phase diagram of Rubisco and EPYC1 concentrations. D values are the average values of 3 repeat experiments. **d** The diffusion coefficients (D) in **c** plotted against overall Rubisco/EPYC1 concentration ratio. Error bars are standard deviations of repeated experiments. Note that because the x -axis is the [Rubisco]/[EPYC1] ratio, in some cases different data points in **c** are clustered into one data point in **d** (for example, D values of 1 μ M Rubisco with 2 μ M EPYC1 and 0.5 μ M Rubisco with 1 μ M EPYC1 are summarized into one data point where the [Rubisco]/[EPYC1] ratio is 0.5). Concentrations are expressed in terms of Rubisco holoenzymes and EPYC1 proteins.”

9. *Materials and Methods, page 21: Please check the plasmid for expressing EPYC1-GFP (pHueFLAGCrEPYC1??)*

The correct description of the plasmid now reads:

“The plasmids expressing EPYC1 and EPYC1-GFP are pHueCrEPYC1 and pHueCrEPYC1-GFP, respectively.”

10. *Legend of supplementary Figure 3: There are no panels d and e in the figure. The x-axis numbering for Rubisco concentration in Fig. S3b is incorrect.*

We thank the reviewer for pointing out these errors. These are fixed in the revised manuscript. The correct Rubisco concentrations in Fig. S3b are now indicated as “10 nM, 100 nM, and 1 μ M”.

Reviewer #2 (Remarks to the Author):

The manuscript by He et. al. describes the role of dilute phase protein complexes in the regulation of protein condensates. By applying the fluorescence correlation spectroscopy (FCS) with a newly developed analytical model, the authors found that the Rubisco and EPYC1 form complexes in the dilute phase no matter if the proteins form condensations or not. Moreover, by analyzing the diffusion coefficients, the authors characterize the composition of the complexes in the dilute phase. This work provides an interesting investigation for understanding how proteins organize in the dilute phase upon co-condensation. To strengthen the work, I have several comments listed below.

We thank the reviewer for the favorable comments.

1. The author applied FCS to determine the binding affinity of Rubisco and EPYC1. However, this result shows $\pm 68\%$ confidence interval, which is relatively low. So, how is the accuracy of the analytical model which is based on the dissociation constant of EYPC1 and Rubisco measured by FCS data.

We note that when comparing the experimental data with the analytical model, there is a free parameter K_b (the dissociation constant between individual EPYC1 and Rubisco stickers), which we adjust to best fit the experimental dilute phase boundaries ($K_b \sim 60\mu\text{M}$ for $K_d = 30\text{ nM}$). As a result, the consistency between the data and the model does not sensitively depend on the measured K_d in the FCS experiment. In practice, keeping the K_b/K_d ratio the same in the model roughly leaves the *dilute-phase* boundary unchanged. The parameters K_b and K_d hardly change the *dense-phase* boundary which depends mainly on the excluded volume interactions.

We have edited the result section on the model to clarify this point:

“Our model uses the experimentally measured dissociation constant of EPYC1 and Rubisco, which from Fig. 2e is $K_d = 29 \pm 12\text{ nM}$ (mean $\pm 65\%$ CI), and then has only one fitting parameter which is the dissociation constant K_b of independent stickers. We fit $K_b = 60\ \mu\text{M}$ to best match the phase diagram measured by experiments. In practice, the precise value of K_d is not critical to the overall agreement between model and data because changing K_b and K_d simultaneously while keeping $K_b^{c_b}/K_d^{\rho_d}$ (see the first terms on the right-hand side of Eqs. 10 and 11 in Materials and Methods) constant roughly leaves the phase diagram unchanged, where ρ_d is the concentration of EPYC1-Rubisco dimers in the dilute phase and c_b is the concentration of bound sticker pairs in the dense phase. However, the parameters K_d and K_b hardly change the dense-phase boundary which depends mainly on the excluded volume interactions.”

2. *Because of the large difference in molecular weight (MW) of Rubisco (550 kDa) and EPYC1 (35 kDa), the author can deduce the complexes' composition from the diffusion coefficients. How about the complexes formed by two different proteins with close MW? Can the FCS method be applied to precisely characterize the composition of the complex?*

The reviewer is correct that this study takes advantage of the large MW difference between EPYC1 and Rubisco. If the two interacting proteins have similar MW, the diffusivity (D) of the heterodimer, considered as a compact sphere, would be only $(1/2)^{1/3} \sim 0.8$ times lower. Differentiating a difference of 20% in D by FCS is challenging. For example, Meseth et al. Biophys. J. 76, 1619 recommends a difference factor of at least 1.6 for two species to be resolvable via FCS (in this study the D ratio between EPYC1 and Rubisco is measured to be ~ 1.7).

We have addressed this in a newly added paragraph in the discussion section. Please refer to our responses to the following point.

3. *As the dilute phase in vitro is homogeneous, the composition of the dilute phase can be determined as shown in this study. Whether this approach can be applied in characterizing the dilute phase in the cellular microenvironment where condensates formed.*

Applying FCS in the cellular context faces with additional challenges. For example, lowered signal-to-noise ratio due to cellular autofluorescence, crowding induced by other biomolecules present, etc. These experiments, while possible, need to be carefully designed and properly analyzed, and are beyond the scope of the current study.

We have added a new paragraph in the discussion section to address the general applicability of the FCS method to study other biomolecular systems, or in the cellular environment.

“In this study, we used fluorescence correlation spectroscopy (FCS) to probe the dilute phase organization of the EPYC1-Rubisco system *in vitro*, in both the one-phase and two-phase regions of the measured phase diagram. FCS has recently been used to probe various properties of biological condensates, including binodals^{27,28} and viscosity in the dense phase²⁷. Here, we use FCS to selectively probe the dilute phase and measure diffusivity of molecules (and molecular complexes) outside of condensates. We then link the measured diffusion coefficient to the presence of EPYC1-Rubisco complexes. Our study takes advantage

of the large difference in molecular weight between EPYC1 (35kDa) and Rubisco (550kDa), so that Rubisco binding to EPYC1 produces a significant slowdown of EPYC1 diffusion (here by a factor of 1.7). We envision the method to be applicable to other systems undergoing phase separation, but care must be taken in the experimental design when the interacting partners have similar sizes²⁹ or when conducting the measurement in a cellular environment²³.”

Reviewer #3 (Remarks to the Author):

Report on Comm Biol Ms by He et al. & Wang, Ms. No.: COMMSBIO-22-2777-T

The work reported in this manuscript on molecular association in the dilute phase in equilibrium with a condensed biomolecular phase is timely and interesting. Using a Rubisco-EPYC1 model system, the experimental results shed new light on the property of such dilute phases, finding Rubisco-EPYC1 dimers in the dilute phase that may compete with condensation. These measurements are highly valuable as they help advance knowledge about the role of dilute-phase properties in biomolecular condensation in general.

We thank the reviewer for highlighting the importance of this work.

However, there are substantial room for improvement in the experimental set up and theoretical modeling--at least in presentation of the results if a more complete study is deemed to be beyond the scope of this manuscript. The issues are related to a lack of measurement of condensed-phase concentration, and therefore a lack of experimental tie-lines (which would be needed for a direct comparison with the theoretical model result in Fig.4a,b). The theoretical model as it stands does not rationalize dimer formation and phase separation using a general physical interaction, but rather accounting for them in two different terms. While some of these shortcomings may be dealt with in future work, their limitations and possible pathways forward should be discussed in the context of closely related works in the literature. More specifically, the following issues should be addressed in revising this manuscript:

1. Is there a rigorous way to translate the experimentally measured diffusion coefficients into EPYC1-Rubisco dimer concentration in the dilute phase? What is the stoichiometry of dilute-phase association? These points are not clear in the manuscript.

We thank the reviewer for this suggestion. We note that in Fig. 4c, we translate the monomer fraction to the “would-be” diffusion coefficient of the EPYC1 monomer-dimer mixture, based on a simple monomer-dimer equilibrium with measured K_d . We could in principle do the same for the experimentally measured D in Fig. 3 and thus convert measured D to a dimer fraction. However, we prefer to show the results of a direct experimental measurement in Fig. 3c (also Fig. 3d) and link the data to our interpretation in Fig. 4c.

We have also revised the Discussion section to better summarize the findings of this work:

“The main finding of this work is the direct experimental observation that EPYC1 and Rubisco form small complexes in both subsaturated solutions and in the dilute phase of a phase-separated, two-component mixture. These complexes seem to contain at most one Rubisco and are most likely EPYC1-Rubisco heterodimers (i.e. 1:1 stoichiometry).”

2. If there is such a correspondence, it will be useful to also show a version of Fig.3c in terms of dimer concentration, not just diffusion coefficients.

Please refer to our reply to the previous point.

3. Why are there more data points in Fig.3c (for $[Rubisco] > 1\mu M$) than in Fig.1d? If there are phase-separated states for $[Rubisco] > 1\mu M$, they should be plotted in Fig.1d as well.

All experiments shown in Fig. 3c were conducted under phase-separated conditions. We have modified Fig. 1d to show the same concentration range as Fig. 3c.

d

4. Because there is no measurement of concentrations of Rubisco and EPYC1 in co-existing condensed phases, there are no experimental tie-lines to compare with the theoretically predicted tie-lines in Fig.4. Condensed-phase protein concentrations have been measured experimentally [see, e.g., Brady et al., Proc Natl Acad Sci USA 114:E8194-E8203 (2017)], and more recently for a two-biomolecule SynGAP-PSD95

model for postsynaptic densities (PSD) [Lin et al., Biophys J 121:157-171 (2022)]. Can these published methods be applied to measure co-existing condensed-phase densities for the authors' EPYC1-Rubisco system?

We thank the reviewer for this comment. We agree that measuring experimental tie-lines and comparing to theoretical models would add additional insights. Specifically, those published methods quoted by the reviewer [based on sedimentation (Brady et al., PNAS 2017) and fluorescence intensity (Lin et al., Biophys J 2022)] should be in principle applicable to the EPYC1-Rubisco system. We have attempted to quantify the dense (and dilute) phase concentration using UV-VIS spectroscopy after centrifugation. However, because EPYC1-Rubisco is a two-component system, quantifying the concentration of both proteins is not as straightforward as for a single-component system (e.g. Ddx4 in Brady et al.). We have also attempted to quantify the dilute phase concentration *in situ*, by analyzing the fluorescence intensity autocorrelation amplitudes. However, a technical hurdle is to quantify the loss of autocorrelation due to non-fluctuating background signal. Nevertheless, we have made progress and hope to publish these results in a future study.

Meanwhile, we note that the main finding of this work is the direct experimental observation that EPYC1-Rubisco forms heterodimers in both subsaturated solutions and in the dilute phase of a phase-separated, two-component mixture. We feel that the experiments presented here, together with a dimer-gel theory with simple approximations (please see our responses to #6-9 below) adequately support our conclusions. Measurements of experimental tie-lines, though potentially insightful, would be better suited for a separate study.

We have expanded the discussion section to explicit mention the potential benefits of tie-line measurements.

“To our knowledge, this work is the first experimental investigation of dilute-phase organization in a biphasic system at equilibrium and our findings pave the way to further experimental and theoretical investigations of how complexes in the dilute phase could regulate condensate stability and dynamics. For example, recent advances in the field have demonstrated many elegant ways to experimentally measure tie-lines^{32,34,35}. These measurements should also be possible for the EPYC1-Rubisco system studied here and would enable more detailed and quantitative testing of theoretical models and simulations^{13,36,37}.”

5. The role of dilute-phase stoichiometric binding in the assembly of biomolecular condensates has been addressed recently using the SynGAP-PSD95 PSD model mentioned above (Biophys J 2022). This work is highly relevant and should be discussed and compared with work reported in the present manuscript. In particular, the PSD model indicates, based on comparison between experimental and theoretical tie-lines

(in several different theoretical scenarios), that auxiliary interactions in addition to dilute-phase stoichiometric binding are needed to account for the assembly of the condensed phase.

We agree. We have now included this reference in the Discussion. Please see our responses below.

6. Indeed, the tie-line pattern exhibited by the current model in Fig.4 corresponds roughly to those shown in Fig.5 in the recent Biophys J 2022 study for the presence of auxiliary interactions – that’s consistent with the current theoretical set-up that, quite artificially (see below), assumes that the physical forces driving dimer formation and forces driving condensation as independent.

We thank the reviewer for highlighting these relevant studies. The Biophys J 2022 and New J Phys 2017 articles are now referenced in the Discussion of our manuscript. Additionally, we have added equations and text to the Materials and Methods section to explain and justify our use of the simplified free-energy model, which is obtained from the full dimer-gel model derived in Ref. 13. We have chosen to present and employ the simplified version of the full model because the simplification reflects the underlying physics of the system: namely, over essentially the entire phase diagram, the model yields either domination by pairing of complete EPYC1 and Rubisco polymers into dimers, or by pairing of independent EPYC1 and Rubisco stickers. Motivated by the reviewer’s comments, we have now substantially revised and expanded the relevant Materials and Methods section (“Analytical model”) to both include the full expression for the dimer-gel theory and to explain and justify our use of the simplifying approximation.

Please refer to the updated manuscript for the revised “Analytical model” section.

7. Possible tie-line patterns (diverging, converging, positive slopes, negative slopes etc.) for two-solute system has been explored in the biomolecular condensates by Lin et al., New J Phys 19:115003 (2017), including how different tie-line patterns for co-mixing and de-mixing are governed by matching/mismatching of charge patterns of a pair of polyampholyte species (Fig.3 of this reference), different Flory-Huggins interaction parameters (Figs.5 and 6 of this reference), and the possibility of ternary phase separation (Fig.7a of this reference). Is ternary phase separation possible for the current study’s EPYC1-Rubisco system? In any case, to put the present work in the context of recent progress in the exploration of two-biomolecular-species phase separation, the scenarios in this earlier New J Phys 2017 work should be discussed.

We agree with the Reviewer that a more detailed consideration of the interactions leading to phase separation in the EPYC1-Rubisco system, including the role of electrostatics, could provide additional insight into the phase diagram such as the pattern of tie lines. While the analyses mentioned above are beyond the scope of the current work, we now emphasize in the Discussion section the importance of these issues for future study based on earlier work in the New J Phys paper. Our experiments did not find conditions of ternary phase separation, and the observation that solutions of only EPYC1 or only Rubisco did not phase separate suggest that a ternary phase is unlikely in this system, but this can also be addressed by varying conditions in future experiments.

Specifically, we have added the following text to the Discussion to better place in context what we have done and how it is related to and contributes to the current body of existing work:

“We further show that a simple analytical model recapitulates all experimental findings. The model includes single EPYC1s and Rubiscos as well as EPYC1-Rubisco heterodimers in the dilute phase but, motivated by the experimental results, neglects higher order complexes such as Rubisco with multiple EPYC1s. The model is also evaluated in mean-field which neglects correlations in the dense phase and includes a minimization over two limits, one where molecular heterodimers dominate and one where independent sticker pairs dominate. Including corrections to this mean-field model could provide additional insight into detailed bonding arrangements in the dense phase as studied for neuronal proteins³². Our experiments do not find ternary phase separation as found in other contexts such as electrostatic models of intrinsically disordered proteins³³. Future studies could include higher order complexes, explicit solvent, and corrections to the mean-field model to more fully characterize both the dense and dilute phases.”

8. *The theoretical model introduced in the current manuscript uses a free energy function with the $\min(F_{dim}, F_{ind})$ term (Eq.1). While this term (together with the other two terms in the free energy function) is useful for providing a qualitative account for the experimental trend (Fig.4a,b), the usage of two different free energies, F_{dim} and F_{ind} , to describe the physical forces responsible for dimer formation on one hand and the physical forces (“sticker-sticker” interactions) for condensation on the other hand means that the experimental data are NOT accounted for by general physical interactions governing BOTH dimer formation AND condensation (that’s how it should work in the real physical world). By doing so, the modeling effort might have missed the opportunity to suggest a possible presence of Rubisco-Rubisco and EPYC1-EPYC1 in the condensed phase (cf. Biophys J 2022 study mentioned above).*

We thank the reviewer for this comment. As described above, we have now included a presentation of the full dimer-gel model for context. In Ref. 13, it was shown for a particular case of associative polymers that the full dimer-gel model is well approximated by the simplified model employed in this paper, and we have confirmed that this also applies to the case of EPYC1 and Rubisco. We now highlight in the Discussion section how the model could be extended, especially regarding the higher order complexes that can form in the dilute and dense phases as mentioned in point 7.

9. Physically, both dimer formation and the “sticker-sticker” contacts must arise from the same general physical interaction. Here, in the authors’ model (Eq.1), the “min” prescription means that there is an abrupt transition from the system being described by F_{dim} to being described by F_{ind} , just when $F_{dim} - F_{ind}$ turns from a very small positive number to a very small negative number, and vice versa. That means the physical forces responsible for dimer formation play no role in condensation in the theoretical model. That’s not reasonable physically. Mathematically, this theoretical set-up might have created an artificial “all-or-none” feature in the model that helps with fitting experimental trend but does not accurately capture the physical interactions involved. These limitations of the theoretical model should be addressed and possible improvements discussed.

We thank the reviewer for raising this question regarding the model. The reviewer is correct that the model separately considers dimer formation from the formation of individual sticker pairs. However, the choice to consider one or the other as dominant is not artificial but rather arises naturally. As we now clarify in the substantially revised Materials and Methods section (“Analytical model”), the full dimer-gel theory from Ref. 13 allows for the coexistence of molecular dimers and individual sticker pairs in both phases, but in practice, for realistic parameters, either one possibility or the other strongly dominates in different regions of the phase diagram. Therefore, to highlight this conceptual simplification, we have explicitly simplified the free energy model via the prescription of employing the minimum of the two limiting free-energy expressions. While we believe this is a physically reasonable minimal model – and follows the established theory of Semenov and Rubenstein for treating sticker-like interactions – the model is only intended as a starting point for interpreting the experimental data. In particular, as a mean-field model, our approach neglects correlations in the dense phase, and also neglects the presence of complexes beyond dimers in the dilute phase. To better acknowledge these limitations, we have now discussed how our model could be improved in future studies (see the revised text quoted under point 7).

REVIEWERS' COMMENTS:

Reviewer #1 (Remarks to the Author):

The authors have satisfactorily addressed my concerns and the manuscript is much improved. Congratulations to the authors.

Reviewer #2 (Remarks to the Author):

The authors addressed my concerns. I support publication of this work in Communications Biology.

Reviewer #3 (Remarks to the Author):

The authors have adequately addressed my previous concerns. I recommend publication of the revised manuscript in Comm Biol.

Typo noted: p.35, 4th line below equation at the top: "To can gain..." should be "To gain ...".